# Eph/Ephrin Promotes the Adhesion of Liver Tissue-Resident Macrophages to a Mimicked Surface of Liver Sinusoidal Endothelial Cells

**DOI:** 10.3390/biomedicines10123234

**Published:** 2022-12-12

**Authors:** Sho Kohara, Kazushige Ogawa

**Affiliations:** 1Laboratory of Veterinary Anatomy, College of Life, Environment and Advanced Sciences, Osaka Prefecture University, 1-58 Rinku-Ourai-Kita, Izumisano, Osaka 598-8531, Japan; 2Laboratory of Veterinary Anatomy, Graduate School of Veterinary Science, Osaka Metropolitan University, 1-58 Rinku-Ourai-Kita, Izumisano, Osaka 598-8531, Japan

**Keywords:** macrophages, Eph, ephrin, ICAM-1, integrin

## Abstract

Kupffer cells are maintained via self-renewal in specific microenvironmental niches, primarily the liver sinusoidal endothelial cells (LSECs). In this study, we propagated tissue-resident macrophages (Mø) from mouse liver using mixed culture with hepatic fibroblastic cells. Propagated liver Mø express *Id3*, *Lxra* and *Spic* transcription factors, which are required for Kupffer cell characterization. Thus, Kupffer cell properties are likely to be maintained in liver Mø propagated using mixed culture with fibroblastic cells. We revealed (i) gene expression of certain Eph receptors and ephrin ligands including EphA2, ephrin-A1, EphB4, and ephrin-B1 in propagated liver Mø and primary LSECs, (ii) immunohistochemical localization of these Eph/ephrin member molecules indicating common expression in Kupffer cells and LSECs, and (iii) surface expression of several integrin α and β subunits, including α4β1, αLβ2, αMβ2, and αXβ2 integrin in propagated liver Mø and that of the corresponding ligands ICAM-1 and VCAM-1 in primary LSECs. Moreover, EphA/ephrin-A and EphB/ephrin-B interactions promoted liver Mø adhesion to the ICAM-1-adsorbed surface, which mimicked that of LSECs and may be implicated in the residence of Kupffer cells in the liver sinusoid. Further studies on regulating the residence and regeneration of Kupffer cells in related hepatic disorders are required to validate our findings.

## 1. Introduction

Macrophages (Mø) are immune cells that function in innate immune responses and are indispensable for the development, maintenance, and regeneration of tissues and organs. In adults, Mø are largely classified into two types: tissue-resident Mø, which colonize tissues/organs at steady state and perform tissue/organ-specific functions to maintain homeostasis; and recruited Mø, which infiltrate lesions in response to tissue/organ damage. Similar to recruited Mø, tissue-resident Mø was previously considered to originate from bone marrow-derived monocytes. However, accumulating evidence has shown that many representative tissue-resident Mø, such as alveolar Mø, Kupffer cells, and microglia in adults originate from fetal precursors, which colonize tissue/organ-specific microenvironments after their migration during embryonic development and differentiate into tissue/organ-specific Mø to perform specific functions. They also undergo self-renewal by local proliferation, in a niche-dependent manner, and persist into adulthood without any infiltration of bone marrow-derived monocytes at a steady state [1,2,3,4,5,6]. Thus, we hypothesized that adult tissue-resident Mø originating from fetal precursors could be propagated alongside the niche-forming cells residing in the respective organ. Therefore, we successfully developed a simple propagation method of tissue-resident Mø using mixed culture with the respective tissue/organ-residing cells to act as niche-forming cells [7]. This method can be commonly applied to the propagation of several tissue-resident Mø.

Kupffer cells are representative tissue-resident Mø residing in hepatic sinusoidal cavities that adhere to liver sinusoidal endothelial cells (LSECs). They are essential for hepatic and systemic homeostasis due to their role in metabolism, scavenging cellular debris and bacteria circulating in the blood, and inducing immunological tolerance [8,9]. Recently, it was demonstrated that the niche of Kupffer cells is composed of LSECs, as well as hepatic stellate cells and hepatocytes [10]. Kupffer cells move within the inside of the sinusoidal cavity along the luminal surface of LSECs to search for cellular debris and bacteria [11]. Unlike other leukocytes, they do not migrate outside the sinusoid into the space of Disse and/or the liver through the bloodstream at a steady state. They can perform specific functions only in their unique niche. However, to the best of our knowledge, the precise mechanism of action of Kupffer cells residing inside the sinusoid remains to be elucidated.

Eph receptors (Ephs), a member of receptor tyrosine kinases, and ephrin ligands (ephrins) are membrane proteins that serve in cell–cell communication systems and play diverse roles in development, physiology, and disease pathogenesis. In mammals, Ephs are divided into EphA (A1–A8 and A10) and EphB (B1–B4 and B6) classes, which bind to the ephrin-A (A1–A5) and -B (B1–B3) classes, respectively. The binding leads to bidirectional signals; forward signals are largely via autophosphorylation and intracellular kinases in receptor-bearing cells, whereas reverse signals are largely via intracellular kinases in ligand-bearing cells. Both signals induce cell migration and cell–cell repulsion by regulating the organization of the actin cytoskeleton mainly through Rho-family GTPases [12,13]. We previously showed that the expression of certain EphAs and ephrin-As was upregulated during monocytic differentiation in both mouse and HL60 monocytes, and stimulation of certain EphAs and ephrin-As promoted cell adhesion likely through interaction with integrins and integrin ligands in HL60 monocytes [14]. Therefore, we speculate that the expression of Ephs and ephrins in Kupffer cells likely regulates their niche. However, to the best of our knowledge, the expression patterns of Ephs and ephrins, as well as their roles in resident Mø, which are restricted to the inside of the sinusoid, have not yet been reported. Therefore, we propagated liver tissue-resident Mø by applying our mixed culture method and examined the expression of Ephs/ephrins and integrins/integrin ligands in propagated liver Mø and primary LSECs. We then examined whether Ephs and ephrins are involved in the adhesion of Kupffer cells to LSECs using propagated liver Mø. Our findings may provide novel insights into the physiology and regeneration of the liver, as specific functions of the Kupffer cell, such as the removal of bacteria circulating in the blood and induction of immunological tolerance, may largely depend on their unique localization in the hepatic sinusoid.

## 2. Material and Methods

### 2.1. Animals

C57BL/6 male mice (n = 24) were obtained from Japan SLC Inc. (Hamamatsu, Japan), and the livers of mice aged 5–6 weeks were used for isolation/propagation of LSECs, and those aged 7–9 weeks were used for propagation/isolation of tissue-resident Mø, as well as for immunohistochemistry and Western blotting experiments. The animal experimentation protocol was approved by the Animal Research Committee of Osaka Prefecture University (approval number: 19-49, 20-32, 22-32). All experiments were performed according to the relevant guidelines of Osaka Prefecture University. Mice were euthanized by intraperitoneal injection of an overdose of pentobarbital (150 mg/kg body weight; 02095-04, Nacalai Tesque Inc., Kyoto, Japan) and intracardially perfused with Hanks′ Balanced Salt Solution (HBSS; Ca/Mg-free, H6648, Sigma-Aldrich, St Louis, MO, USA) including 50 U/mL heparin (224122485, Mochida Pharmaceutical, Tokyo, Japan) to remove the blood cells. Then, the aseptically dissected liver was immediately dipped in ice-cold HBSS.

### 2.2. Propagation of Tissue-Resident Macrophages Using Mixed Culture with Liver Cells

Liver cells, including Kupffer cells, were mixed-cultured according to the method described by Ogawa et al. with modifications [7]. Briefly, after removing the gallbladder, approximately half the volume of the liver was minced with a razor blade into approximately 1 mm^3^ pieces and put in 15 mL conical tubes containing 12.5 mL collagenase solution: 0.5 mg/mL collagenase (C5138, Type IV, Sigma-Aldrich) and 1 mM CaCl_2_ (Col/HBSS) in 20 mM HEPES-buffered HBSS (pH 7.4). Then, the liver tissues were digested for 50–60 min at 37 °C with gentle stirring at 120 rpm with one change to the collagenase solution. After washing with HBSS, undigested tissues were further dispersed by pipetting. The suspensions were sedimented for 5 min at 100× *g* (Model 2410, Kubota, Tokyo, Japan). Then, the cells/tissues were put in three 10 cm tissue culture dishes (3020-100, AGC Techno Glass, Haibara, Japan) and cultured with 10% fetal bovine serum (FBS; 175012, Nichirei Biosciences Inc., Tokyo, Japan) and 100 U/mL penicillin/100 μg/mL streptomycin (pen/strep; P4333, Sigma-Aldrich) in DMEM (D6046, Sigma-Aldrich; FBS/DMEM). The medium was replaced every 3 days until the dish surfaces were covered by multilayered cells largely composed of Mø and other non-parenchymal cells showing fibroblastic morphology. Over-confluent cells were detached with 0.1% trypsin plus 2 mM EDTA in HBSS (Tryp/EDTA/HBSS), and then by pipetting. Subsequently, the cells at a dilution ratio of 1:3 were subcultured/maintained in the same medium until they became over-confluent again or frozen at −80 °C in a cell suspension with Bambanker (CS-02-001, Nippon Genetics, Tokyo, Japan) as a cryopreservative. 

### 2.3. Separation of Liver Tissue-Resident Macrophages Propagated Using Mixed Culture from Fibroblastic Cells

Mixed-cultured liver Mø were separated from non-parenchymal fibroblastic cells according to the method described by Ogawa et al. [7]. Briefly, mixed-cultured, over-confluent cells up to 4 passages (usually 1–2 passages) were used for the purification of liver Mø. Over-confluent cells collected from a 10 cm tissue culture dish were seeded/cultured in a 10 cm (1-7484-01, As One, Osaka, Japan) or 5.5 cm bacteriological Petri dish (1-8549-02; As One) containing FBS/DMEM. At 16 h after seeding, fibroblastic cells usually formed nonadherent cell aggregates in the dish and the Mø selectively adhered to the dish surface. The adherent cells that were washed with conditioned media to remove nonadherent cells were used for reverse transcription-polymerase chain reaction (RT-PCR). For flow cytometry and phagocytosis assay as well as cell adhesion stripe assay, adherent cells (Mø) were detached using 5 mM EDTA in HBSS. To remove the cell aggregates, the cell suspension was passed through a 40 µm cell strainer (352235, BD Falcon, Franklin Lakes, NJ, USA), and sedimented at 220× *g* for 5 min. Then, the cells were resuspended in phosphate-buffered saline (PBS; Ca/Mg-free, 1102P10, Cell Science & Technology Institute, Inc., Yamagata, Japan) containing 1% bovine serum albumin (BSA; A3059, Sigma-Aldrich), 2 mM EDTA, and 0.01% NaN_3_ (BSA/EDTA/PBS) for flow cytometry or in HBSS for phagocytosis assay and cell adhesion stripe assay. The number of cells was then calculated and used for further experiments.

### 2.4. Separation of Fibroblastic Cells Propagated Using Mixed Culture from Liver Tissue-Resident Macrophages 

As nonadherent cell aggregates in a bacteriological Petri dish included liver tissue-resident Mø as a minor population, non-parenchymal fibroblastic cells propagated in mixed culture with the Mø were purified from the cell aggregates. The culture media containing nonadherent cells/aggregates in a bacteriological Petri dish were centrifuged for 5 min at 100× *g* and incubated in Tryp/EDTA/HBSS for 5 min at 37 °C. Then, the cells were centrifuged again and plated/cultured in a 10 cm tissue culture dish with FBS/DMEM. On the day after plating, the cells were treated with 2×-diluted Tryp/EDTA/HBSS for approximately 3 min at 37 °C when the non-parenchymal fibroblastic cells except liver Mø were mostly detached from the dish (most liver Mø remained attached to the dish). The detached cells were again harvested, centrifuged, and cultured in a tissue culture dish with FBS/DMEM. These procedures were repeated at least once to purify non-parenchymal fibroblastic cells by removing strongly adhesive liver Mø. Subsequently, the cells were used for RT-PCR analyses.

### 2.5. Collection and Propagation of LSECs

LSECs were isolated according to the method described by Cabral et al. with some modifications [15]. After clearing blood cells using intracardial perfusion, minced liver tissues (~1 mm^3^ pieces per mouse) were digested with 12.5 mL Col/HBSS in 15 mL conical tubes with gentle stirring at 120 rpm for 30 min at 37 °C. After washing cell/tissue suspensions with HBSS, they were further dispersed by pipetting. To remove cell aggregates, the suspensions were passed through a 100 µm cell strainer (352360, BD Falcon) and then centrifuged at 4 °C for 3 min at 100× *g* (LX-120, Tomy, Tokyo, Japan). The supernatant, which is a non-parenchymal cell-rich fraction, was collected for further separation of the LSECs. Because LSECs were also included in the sediment cells/tissues, they were resuspended in HBSS, passed through a 40 µm cell strainer (352235, BD Falcon), and again centrifuged at 100× *g* for 3 min at 4 °C. Then, the supernatant together with the supernatant obtained from the first round of centrifugation was sedimented at 180× *g* for 8 min at 4 °C. The precipitates were resuspended in RPMI-1640 (R8758, Sigma-Aldrich) and centrifuged at 30× *g* for 2 min at 4 °C to remove hepatocytes. Then the supernatant was collected as a non-parenchymal cell fraction that was mostly composed of LSECs and Mø and sedimented at 180× *g* for 8 min at 4 °C. The sedimented cells were plated on a 10 cm bacteriological Petri dish containing RPMI-1640 supplemented with 10% FBS and penicillin/streptomycin and incubated at 22 °C for 5 min to allow liver Mø to adhere to the dish surface. Nonadherent cells in the medium were collected, sedimented at 260× *g* for 5 min, and seeded in four 6 cm tissue culture dishes (3020-100, AGC Techno Glass) that were coated with collagen (1.0 mL of 50 µg/mL collagen per dish at 37 °C for 2 h; Type I-C, Cellmatrix, Nitta Gelatin Inc., Osaka, Japan), and cultured in a dedicated medium for microvascular endothelial cells, EBM-2MV (CC-3202, Lonza, Basel, Switzerland). The medium was refreshed daily until use in further experiments.

### 2.6. Phagocytosis Analysis with Fluorescent Beads

The phagocytic properties of liver tissue-resident Mø propagated using mixed culture were examined according to the method described by Ogawa et al. [7]. After separation of the Mø on the Petri dish, the cells (2.5 × 10^5^/0.5 mL FBS/DMEM) were put in a 5 mL siliconized tube (Siliconize L-25, 0411002, Fuji-Rika Industries, Osaka, Japan) to prevent adhesion to the tube wall. Then, 1.0 µL fluorescent yellow–green-conjugated latex beads (mean diameter, 1 µm; L4655, Sigma-Aldrich) were added to the cell suspension. The cells were then incubated at 37 °C for 1 h with gentle shaking at 18 rpm on a seesaw-type shaker (Wave SI Slim; Taitec, Koshigaya, Japan), washed thrice with HBSS, and plated on a 3.5 cm glass-bottom dish (3910-035, AGC Techno Glass) with 1.5 mL FBS/DMEM for approximately 60 min until almost all the cells adhered to the glass surface. Thereafter, the cells were fixed with 10% formalin (Kanto Chemical, Tokyo, Japan) in PBS (Formalin/PBS) for 10 min. Green fluorescence and phase-contrast images of the same field were captured under a fluorescence microscope (IX71; Olympus, Tokyo, Japan) using a digital camera (DP72; Olympus) controlled by the manufacturer’s software (DP2-BSW; Olympus). The cells that phagocytosed more than two latex beads were denoted as Mø. More than 690 cells per sample were counted, and the percentage of Mø in each mouse was calculated from three independent experiments involving three mice. Data are presented as mean ± SD.

### 2.7. Total RNA Extraction and RT-PCR Analyses

We isolated total RNA from the propagated liver Mø, LSECs, and non-parenchymal fibroblastic cells using TRI Reagent (TR118, Molecular Research Centre, Inc., Cincinnati, OH, USA), and performed RT-PCR analysis according to the method of Mukai et al. [14]. In brief, according to the manufacturer’s instructions, 1.0 µg of total RNA was transcribed into first-strand cDNA using RNase H^−^ reverse transcriptase (316-08151, Nippon Gene, Toyama, Japan) and oligo (dT)_18_ primers. To detect the expression of transcripts using reverse-transcribed cDNA as the template, 0.5 µL of a 25 µL reaction mixture was amplified with Taq DNA polymerase (TaKaRa Ex Taq HS, RR006A; TaKaRa Bio Inc., Otsu, Japan). The primer pairs, annealing temperature, and cycling number used for PCR amplification are listed in Appendix A. The PCR products were separated on 1.5% agarose gels and visualized using ethidium bromide staining. As a negative control, the RT reaction was excluded. Transcript expression was determined from more than three independent experiments in liver Mø, fibroblastic cells, and LSECs propagated/segregated from liver cells derived from more than three mice.

### 2.8. Flow Cytometry

Flow cytometry was performed to examine the expression of Mø/monocytic markers and integrins in tissue-resident Mø propagated using mixed culture, as well as to determine the expression of the marker of LSECs and integrin ligands in segregated LSECs. The monoclonal antibodies used in flow cytometric analyses are listed in Appendix A. Cells at a concentration of approximately 1 × 10^6^ cells/0.5 mL in BSA/EDTA/PBS were fixed with 5% formalin in BSA/EDTA/PBS for approximately 20 min at 24 °C. After washing with BSA/EDTA/PBS, the cells were permeabilized with 0.2% saponin (Nacalai Tesque, Kyoto, Japan) in 1 mL BSA/EDTA/PBS (Sap/BSA/EDTA/PBS) for 10 min at 24 °C. To detect integrin ligands, we used LSECs without fixation. To avoid non-specific Fc-gamma receptor-mediated binding of fluorochrome-conjugated antibodies, the cell suspensions (~4 × 10^5^ cells/50 µL for liver Mø; ~1.2 × 10^5^ cells/50 µL for LSECs) were pre-treated with 0.5 µg of anti-mouse CD16/32 antibody for 10 min at 24 °C. Following the manufacturer’s instructions, we added suitable amounts of fluorochrome-labeled test antibodies (listed in Appendix A) to the 50 µL cell suspension, followed by incubation for 20 min at 4 °C. After washing with Sap/BSA/EDTA/PBS or BSA/EDTA/PBS, 20,000 cells were analyzed for their expression characteristics using flow cytometry (S3 Cell Sorter, Bio-Rad Laboratories, Hercules, CA, USA; CytoFLEX S, Beckman Coulter, Brea, CA, USA). Cell suspensions pre-treated with the anti-mouse CD16/32 antibody were used as controls and further treated with the same fluorochrome-labeled isotype control antibody of the same amount as the test antibody. The expression of molecules was determined in more than three independent experiments in cells (Mø and LSECs) propagated from liver tissues derived from more than three mice.

### 2.9. Immunofluorescence Staining

Immunofluorescence staining was performed to determine the histological localization of EphA2, ephrin-A1, EphB4, and ephrin-B1 in the liver. The liver tissues cut to 2–3 mm thickness were fixed with Formalin/PBS for approximately 3 h at 4 °C. After washing with PBS, the tissues were immersed in 30% sucrose in PBS overnight and mounted in an optimum cutting temperature compound (Sakura Finetechnical Co., Ltd., Tokyo, Japan). Next, 4 µm thick cryostat sections were used for fluorescence staining. Goat polyclonal antibodies against EphA2 (AF639), EphB4 (AF446), and ephrin-B1 (AF473) (R&D Systems, Minneapolis, MN, USA) and rabbit polyclonal antibody against ephrin-A1 (SAB4500696; Sigma-Aldrich) were used. Rat monoclonal antibodies against mouse CD146 (cell surface glycoprotein MUC18; ME-9F1, Miltenyi Biotec, Bergisch Gladbach, Germany), and mouse monocyte/macrophage antigen F4/80 (BM8, BMA Biomedicals, Augst, Switzerland) were used to label LSECs and Kupffer cells, respectively. Double immunofluorescence staining was performed. Cryostat sections were incubated in a humid chamber with BSA/PBS, followed by incubation with a mixture of primary antibodies at a concentration of 5 µg/mL (anti-EphA2, anti-ephrin-A1, anti-EphB4, anti-ephrin-B1, anti-CD146) or 2 µg/mL (anti-F4/80) for 1.5 h at 32 °C. After washing with PBS, the sections were incubated with a mixture of Alexa Fluor 488-conjugated donkey anti-rat IgG (5 µg/mL; Molecular Probes, Inc., Eugene, OR, USA) and Alexa Fluor 594-conjugated donkey anti-goat IgG (5 µg/mL; Molecular Probes) or a mixture of Alexa Fluor 488-conjugated donkey anti-rat IgG and Alexa Fluor 594-conjugated donkey anti-rabbit IgG (5 µg/mL; Molecular Probes) for 30 min at 32 °C. The sections were then washed with PBS, mounted with PermaFluor (K002; Thermo Fisher Scientific, Waltham, MA, USA), and photographed under a fluorescence microscope (IX71; Olympus). Sections were stained with 4ʹ,6-diamidino-2-phenylindole dihydrochloride (DAPI, 2 µg/mL; Wako Pure Chemical Industries, Ltd., Osaka, Japan), which was included in the secondary antibody mixture. The specificity of the staining was verified by incubation without primary antibodies. The immunohistochemical localization of those molecules was determined from three independent experiments using 42 liver sections derived from three mice.

### 2.10. Immunoprecipitation and Immunoblotting

To examine tyrosine phosphorylation of endogenous Ephs in the liver, we performed immunoblotting using immunoprecipitated samples. Adult mouse livers were homogenized in modified RIPA buffer (1% sodium deoxycholate, 0.1% SDS, 1% Triton X-100, 1 mM EDTA, 1.5 mM MgCl_2_, 150 mM NaCl, 100 mM sodium fluoride, 10 mM sodium pyrophosphate, and 10% glycerol in 50 mM Tris-HCl pH 7.5) containing protease inhibitors (10 µg/mL aprotinin, 10 µg/mL leupeptin, 2 µg/mL pepstatin-A, and 0.5 mM phenylmethylsulfonyl fluoride) and a phosphatase inhibitor (1 mM sodium orthovanadate). Supernatants were collected after high-speed centrifugation and protein concentrations were measured using a Protein Assay kit (Bio-Rad Laboratories, Hercules, CA, USA).

For immunoprecipitation, 800 µg of tissue extracts were incubated for 16 h at 4 °C with 4 µg anti-EphA2 (rabbit IgG; sc-924, Santa Cruz Biotechnology, Santa Cruz, CA, USA) or 0.5 µL normal rabbit serum (as a control; S-5000-20, Vector Laboratories, Inc., Burlingame, CA, USA), and with 4 µg anti-EphB4 (goat IgG; R&D Systems) or 0.5 µL normal goat serum (S-1000-20, Vector), followed by treatment with 15 µL protein G magnetic beads (DB10003, Thermo Fisher) for 1 h at 4 °C. The immunoprecipitates were separated on 10% polyacrylamide gels and transferred onto polyvinylidene fluoride membranes, which were incubated overnight at 4 °C in Tris-buffered saline with 0.1% Triton X-100 (TBS-T) containing 3% BSA and 1:5000 HRP-conjugated PY20 (610012, BD, Franklin Lakes, NJ, USA). Immunoblots were developed using ECL Prime chemiluminescence reagents (10308449, Thermo Fisher). The membrane was reprobed with 1.0 µg/mL anti-EphA2 or 0.2 µg/mL anti-EphB4 antibody in TBS-T containing 3% BSA and 0.2% non-fat dry milk. After incubation with 1:5000 HRP-conjugated Protein A (55901, MP Biomedicals, Solon, OH, USA) or 1:20,000 HRP-conjugated mouse anti-goat IgG (205-035-108, Jackson ImmunoResearch Laboratories, Inc., West Grove, PA, USA), immunoblots were redeveloped.

### 2.11. Cell-Adhesion Stripe Assay

We examined the adhesion of liver Mø to ICAM-1-coated coverslips mimicking the surface of LSECs on which the ephrin or Eph protein was adsorbed in stripes, according to the method of Mukai et al. [14] with modifications. Briefly, coverslips (diameter, 15 mm) were incubated overnight with poly-L-ornithine in PBS (100 µL of 100 µg/mL; P2533, Sigma-Aldrich), washed with sterile water, and dried. Comb-shaped silicon masks containing parallel teeth and gaps (each approximately 0.48 mm wide) were applied on the glass surface, and then 100 µL of 10 µg/mL ephrin-A1-Fc (602-A1, R&D Systems), EphA2-Fc (639-A2, R&D Systems), ephrin-B1-Fc (473-EB, R&D System), EphB4-Fc (446-E4, R&D System), or human IgG Fc as a control (Fc; 55911, MP Biomedicals) were adsorbed onto the surface at 37 °C for 60 min. The coverslips were then washed with PBS and the masks were removed. The coverslips were washed again, and then ICAM1-Fc (100 µL of 10 µg/mL in HBSS; 796-IC, R&D Systems) was adsorbed on the surface at 37 °C for 60 min. After washing, five coverslips with different ephrin/Eph stripes were placed in a 6 cm diameter culture dish containing 4 mL of FBS/DMEM. Next, propagated liver Mø (7.5 × 10^5^ cells) were plated and allowed to adhere for 0.5 h at 24 °C and then for 15.5 h at 37 °C. Cells on the coverslips were then fixed with Formalin/PBS for 20 min at 24 °C. 

The cells on the coverslips were washed with PBS and incubated with 0.02% TritonX-100 in PBS for 10 min at 24 °C. To visualize the actin filaments, cells were pre-incubated in a humid chamber with BSA/PBS for 30 min, followed by incubation with a mixture of 165 nM Alexa 568-labeled phalloidin (A12380, Thermo Fisher) and 2 µg/mL DAPI in BSA/PBS for 30 min at 32 °C, followed by washing with PBS and mounting with PermaFluor (Thermo Fisher). Phase-contrast images, orange-red fluorescence images (actin), and blue fluorescence images (nucleus) were captured from the same fields, including regions coated with ephrin-A1-Fc, EphA2-Fc, ephrin-B1-Fc, EphB4-Fc, or Fc plus the integrin ligand (test regions), and the adjacent control regions coated with the integrin ligand alone, using a 4× objective lens and a 10× objective lens (IX71; Olympus). To compare cell densities in the test and control regions, we counted cells in the same field (area, ~0.48 × ~2.32 mm) in each test and control region, respectively. The relative cell densities were determined from three independent experiments and normalized relative to those determined for the integrin-ligand-coated control region.

### 2.12. Statistical Analyses

Statistical analyses were performed using Microsoft Excel and statistical software (http://statpages.info/anova1sm.html; accessed on 30 September 2022). Differences in relative cell densities in the test regions were compared to those in the adjacent control region among the five different coated proteins and were evaluated using one-way analysis of variance, followed by Tukey’s HSD posthoc analysis. *p*-values less than 0.05 were considered significant. All values represent the mean ± SD.

## 3. Results

### 3.1. Propagation Behavior of Liver Mø in Mixed Culture and Their Segregation

Tissue-resident Mø were propagated in mixed culture with non-parenchymal cells from the mouse liver. Liver Mø showed high propagation in FBS/DMEM without additional Mø growth factors such as CSF1. Primary hepatic non-parenchymal cells, including Mø, proliferated, reached over-confluence, and formed a multilayered structure on a standard tissue culture dish within 7–14 days, with the culture medium changed every three days (Figure 1A,B). These cells were then subcultured at a dilution ratio of 1:3 until they reached over-confluence again, which occurred within a similar period (Figure 1C−F). Most primary hepatocytes were not cultivated in a tissue culture dish with a standard culture medium. After the first passage, no clear large epithelial cells (hepatocytes) appeared in the mixed culture, and the propagated cells consisted mostly of Mø and fibroblastic cells (Figure 1C,E). Mø were morphologically identified as small cells, specifically small flat cells with a few cytoplasmic protrusions and small round or fusiform cells in the mixed culture (Figure 1C,D). The non-parenchymal cells, including Mø, were sub-cultured for more than six passages, showing similar propagation behavior in the mixed culture as observed in the primary cells. We usually used the over-confluent cells within 1–2 passages for subsequent analyses. The frozen cells were found to propagate similarly to the unfrozen cells when over-confluent cells were frozen at a dilution ratio of 1:3 then thawed, and cultured under the same cultivation conditions.

Based on their different adhesive properties to the bacteriological Petri dish, Mø were separated from the other non-parenchymal cells (fibroblastic cells) in the mixed culture since only Mø adhere to the dish. After several hours of seeding over-confluent cells, small round/fusiform cells with a few cytoplasmic protrusions, identified as Mø, adhered to the dish surface, with evidence of cell aggregates floating in the media. These cell aggregates were easily eliminated by washing with the conditioned media (Figure 1G,H). We generally collected approximately 2−3 × 10^6^ adherent cells per over-confluent cell harvested from a 10 cm tissue culture dish and analyzed them for typical Mø-specific features.

We precisely determined the percentage of Mø in segregated cells that adhered to bacteriological Petri dishes by evaluating the phagocytosis of fluorescent beads. Almost all cells which adhered to the Petri dish phagocytosed the fluorescent beads during incubation, and most cells contained numerous beads in their cytoplasm, highlighting their increased phagocytotic ability (Figure 2A). The bead-positive and bead-negative cells were examined to estimate the percentage of Mø among the segregated cells. We defined cells phagocytosing more than two beads as bead-positive cells and counted more than 690 cells per sample. Overall, these cells comprised 98.3 ± 0.1% of the entire population (Figure 2B). Thus, the Mø segregation method using the adhesion property of the bacteriological Petri dish is a reliable and simple purification method to obtain liver tissue-resident Mø from other non-parenchymal cells in mixed culture.

### 3.2. Mø Marker Expression Profiles Using RT-PCR and Flow Cytometry in Propagated Liver Mø

Certain transcription factors are specific for and representative of tissue-resident Mø [1,2]. Further, ID3, SPIC, and LXRα expression has been shown as a characteristic feature of Kupffer cells [6,10]. Thus, we examined the expression of these transcription factors using RT-PCR and found that *Id3*, *Spic*, and *Lxra* were clearly expressed in Mø propagated in the mixed culture with fibroblastic cells (Figure 2C). We also examined the expression profiles of nine Mø markers in propagated Mø using flow cytometry. Propagated Mø showed high expression of CD36 and CD169, moderate/low expression of CD68, CD206, arginase 1, F4/80, and Mertk, and almost no expression of MHC II and iNOS (Figure 2D). Except for arginase 1, liver Mø were composed of a single fraction based on the histograms of expression patterns of the markers. The positive expression of these transcription factors revealed that Mø likely maintain the properties of Kupffer cells and undergo polarization to the M2 phenotype in the mixed culture.

### 3.3. Identification of Segregated LSECs and Fibroblastic Cells Propagated Using Mixed Culture with Mø

We propagated LSECs with a dedicated medium for microvascular endothelial cells after the collection of hepatic non-parenchymal cells using centrifugation and segregation of Mø (Figure 3A). CD146 is an LSEC-specific marker, and the propagated cells showed morphologically small epithelial cell features, were CD146-positive, and CD11b-negative (Figure 3A), leading to the successful segregation/propagation of LSECs [16], which were subsequently employed for expression analyses.

Liver tissue-resident Mø propagated along with fibroblastic cells in the mixed culture. We hypothesized that fibroblastic cells function as a niche for nursing liver Mø in vitro; thus, we segregated them to examine their properties as a niche, which has previously been shown to be comprised of LSECs, hepatic stellate cells, and hepatocytes [10]. The cells other than Mø in the mixed culture showed fibroblastic morphologies and properties of *Desmin*-positive cells, but were *Gfap*- and *Ngfr*-negative (Figure 3B), all of which are markers of hepatic stellate cells [17,18]. We then examined the expression of essential molecules as niche properties to maintain tissue-resident Mø and promote their proliferation. We found that both cultured LSECs and fibroblastic cells expressed *Csf1* and *Il34*, but not *Csf2* (Figure 4A), and these results are almost identical to previous findings [6,10]. We also found that propagated liver Mø expressed not only *Csf1r and Csf2ra*, but also CD115 (CSF1R) and CD116 (CSF2RA) (Figure 4B). Thus, CSF1/CSF1R interaction likely induced the proliferation of Mø in mixed cultures, as shown in a previous report [19].

### 3.4. Integrin/Integrin Ligand and Chemokine Receptor/Chemokine Expression in Propagated Liver Mø, Fibroblastic Cells, and LSECs

We speculated that the adhesion of Kupffer cells to LSECs via the integrin/integrin ligands is inevitable due to their presence in the sinusoidal luminal space. Thus, we screened the propagated liver Mø for the expression of certain integrins that are generally expressed in leukocytes [14,20]. We found that Mø expressed *Itga4*, *Itga5*, *Itga6*, *Itgal*, *Itgam*, *Itgax*, *Itgb1*, and *Itgb2*, but not *Itga1*, *Itga2*, *Itga3*, and *Itgad* using RT-PCR (Figure 5A), and we confirmed the expression of these proteins in Mø using flow cytometry analysis (Figure 5B). These findings indicated that liver Mø likely express α4β1, α5β1, α6β1, αLβ2, αMβ2, and αXβ2 integrins. Because integrins α4β1, αLβ2, αMβ2, and αXβ2 possess binding properties with ICAM-1 and/or VCAM-1, which are major integrin ligands expressed as membrane proteins in vascular endothelial cells [21,22], we screened LSECs as well as fibroblastic cells propagated in vitro for ligand expression. We found clear *Icam-1* and *Vcam-1* expression in these cells using RT-PCR, strong ICAM-1, and substantial VCAM-1 expression in LSECs using flow cytometry (Figure 6A,B). Thus, these integrins and integrin ligands are possibly involved in cell-cell adhesion between Kupffer cells and LSECs.

Chemokines rapidly induce integrin activation in leukocytes [23], and previous studies have shown that LSECs and hepatic stellate cells express chemokines CCL2, CXCL9, and CXCL12, and Kupffer cells express chemokine receptors CCR2 and CXCR3 [24]; CXCL12/CXCR4 regulates the function or differentiation of monocytes/Mø [25]. Thus, we examined chemokine expression in propagated LSECs and fibroblastic cells, as well as the expression of chemokine receptors in propagated liver Mø. We found *Ccl2* and *Cxcl12* expression in both the LSECs and fibroblastic cells, as well as *Ccr2*, *Cxcr3*, and *Cxcr4* expression in the liver Mø (Figure 7A,B). Moreover, we found clear protein expression of CD184 (CXCR4) and CD192 (CCR2) in propagated Mø. Thus, CCL2/CCR2 and CXCL12/CXCR4 may be involved in cell-cell adhesion between liver Kupffer cells and LSECs, mediated by the interaction of integrins and integrin ligands.

### 3.5. Localization of EphA2, Ephrin-A1, EphB4, and Ephrin-B1 in the Liver

Our previous studies showed that EphA2 and ephrin-A1 induce adhesion in monocytes via integrin/integrin ligands [14]; EphB4 and ephrin-B1 induce cell-cell repulsion in gut epithelial cells [26]; EphA2, ephrin-A1, EphB4, and ephrin-B1 are expressed in vascular endothelial cells [27,28]. Thus, we performed immunohistochemistry analysis to examine EphA2, ephrin-A1, EphB4, and ephrin-B1 localization in the mouse liver using CD146 as a marker of LSECs and F4/80 as a marker of Kupffer cells. We found that EphA2, ephrin-A1, EphB4, and ephrin-B1 were localized in CD146-positive LSECs (Figure 8). Further, ephrin-B1 immunoreactivity was stronger in LSECs distributed in the marginal region than in the middle/central region of hepatic lobules, whereas a vice versa EphB4 immunoreactivity was observed in LSECs. Moreover, EphA2, ephrin-A1, EphB4, and ephrin-B1 immunoreactivity was weak but substantial in F4/80-positive Kupffer cells (Figure 8). Furthermore, we found a strong and weak tyrosine-phosphorylation of EphA2 and EphB4 proteins, respectively, in the liver (Figure 9A). These findings indicate that EphA2-bearing and EphB4-bearing cells likely come into contact with ephrin-A1-bearing and ephrin-B1-bearing cells, respectively, and are activated in the liver.

### 3.6. Expression of Eph and Ephrin-A1 in Propagated Liver Mø and LSECs

Certain Ephs and ephrins are expressed in Kupffer cells and LSECs, therefore, we screened propagated liver Mø and LSECs for the expression of all members of Ephs and ephrins using RT-PCR. We found that the propagated liver Mø clearly expressed *Epha2*, *Epha4*, *Efna1*, *Efna4*, *Efna5*, *Ephb3*, *Ephb4*, *Ephb6*, *Efnb1*, *Efnb2*, and *Efnb3* (Figure 9B), whereas segregated LSECs clearly expressed *Epha2*, *Efna1*, *Ephb4*, *Efnb1*, *Efnb2*, and *Efnb3* (Figure 9C). These findings indicate that both EphA/ephrin-A and EphB/ephrin-B interactions likely arise in Kupffer cells when they come into contact with LSECs as the members of the EphA and EphB receptor classes promiscuously bind to ligands of the ephrin-A and ephrin-B (B1–B3) classes, respectively [29].

### 3.7. EphA/Ephrin-A, EphB/Ephrin-B Activation Promotes Adhesion to ICAM-1-Coated Surface in Liver Mø

By using the mimicked surface of LSECs and propagated liver Mø, we explored whether EphA/ephrin-A and EphB/ephrin-B activation affect integrin-mediated cell adhesion of Kupffer cells to the luminal surface of LSECs. Thus, we compared the adhesive properties of propagated liver Mø on coverslip surfaces with stripes of ephrin-A1-Fc, EphA2-Fc, ephrin-B1-Fc, EphB4-Fc, or Fc (control protein). The test stripes were composed of a test region of both the ephrin-/Eph- and ICAM-1-protein adsorbed surface and a control region of ICAM-1-protein adsorbed surface alternately presented in stripes at certain intervals on the coverslip (Figure 10A). Using phase-contrast and DAPI-stained micrographs, we found that the Mø formed stripes with different cell densities corresponding to the test and control surfaces, preferentially occupying ICAM-1- and ephrin-A1-, EphA2-, ephrin-B1-, or EphB4-adsorbed test surfaces rather than adjacent ICAM-1-adsorbed control surfaces (Figure 10B). In contrast, liver Mø did not form stripes of different cell densities on the surface of control stripes, in which Fc was adsorbed instead of the Fc-chimera protein. We calculated the cell densities in the Fc-chimera protein-adsorbed test region relative to those in the adjacent control region. The relative cell densities on the ephrin-A1-Fc-, EphA2-Fc-, ephrin-B1-Fc-, EphB4-Fc-, and Fc-adsorbed regions compared to the control region were 2.30 ± 0.32, 2.28 ± 0.21, 1.75 ± 0.07, 1.63 ± 0.11, and 0.92 ± 0.11 folds (mean ± SD), respectively (Figure 10C). The values of the test stripes were significantly different from those of the control strips (ephrin-A1-Fc, *p <* 0.001; EphA2-Fc, *p <* 0.001; ephrin-B1-Fc, *p =* 0.005; and EphB4-Fc, *p =* 0.015). These results indicate that liver Mø preferentially reside on the Eph/ephrin-bearing cell surface, where the membrane-type integrin ligand is also likely expressed in LSECs at a steady state. By actin staining with these stripes to clearly observe the cell margin, we easily identified that a blank zone without cells appeared in the control region adjacent to the ephrin/Eph protein-adsorbed test region and that boundaries between the control and test strips were much clearer in the ephrin-A1 and EphA2 stripes than in the ephrin-B1 and EphB4 stripes (Figure 10D).

## 4. Discussion

### 4.1. Propagation of Liver Tissue-Resident Mø in Mixed Culture

Many representative tissue-resident Mø, including Kupffer cells, originate from fetal precursors, which colonize tissue/organ-specific niches during embryonic development, differentiate into tissue/organ-specific Mø, and proliferate locally by receiving adequate nourishment from the niches and persist into adulthood, without any infiltration of bone marrow-derived monocytes at a steady state [1,2,5,6,10]. LSECs provide an indispensable niche for adherent Kupffer cells [5,10], and hepatic stellate cells and hepatocytes have also been shown to share a niche that characterizes the properties of Kupffer cells [6,10]. We successfully propagated tissue-resident Mø from the C57BL/6 mouse liver by applying our method of mixed culture with standard culture media [7]. In the mixed culture of liver cells, tissue-resident Mø propagated similarly to hepatic non-parenchymal cells showing fibroblastic morphology. We found that the fibroblastic cells expressed (i) *Csf1* and *Il34*, both of which are trophic factors that generally induce the proliferation of tissue-resident Mø, including Kupffer cells [2,6], and (ii) *Ccl2* and *Vcam-1*, which are indispensable molecules in the niche of Kupffer cells [10]. Thus, by the expression of these molecules, fibroblastic cells likely function as a niche for the proliferation and characterization of liver tissue-resident Mø in the mixed culture. We also found that Mø propagated in mixed culture with fibroblastic cells clearly expressed *Id3*, *Lxra*, and *Spic* transcription factors, which are important for characterizing Kupffer cells [1,2,10]. Moreover, expression analyses using flow cytometry showed that the expression patterns of Mø markers in propagated Mø were largely similar to those in Kupffer cells of the same mouse strain ex vivo [30]. Thus, we speculate that the properties of Kupffer cells could be maintained in the liver Mø propagated using mixed culture with only hepatic fibroblastic cells, and/or, within a few passages of the subculture, Kupffer cells may not lose their essential properties by mixed culturing without LSECs serving their niche as a main component. Based on their morphology and expression of molecules related to the niche, such as *Csf1*, *Il34*, *Icam-1*, *Vcam-1*, *Ccl2,* and *Cxcl12*, we presumed that fibroblastic cells propagating in the mixed culture were likely derived from hepatic stellate cells. However, the fibroblastic cells propagated using mixed culture with liver Mø expressed only *Desmin* but not *Gfap* and *Ngfr*, all of which are representative markers for identifying hepatic stellate cells [17,18]. Thus, further studies are required to determine the nature of fibroblastic cells. Chronic inflammation in the liver leads to fibrosis, in which Kupffer cells and recruited Mø are involved in the activation of hepatic stellate cells and fibrosis progression [31,32]. However, the roles of Kupffer cells discriminated from those of recruited Mø during fibrosis progression are not fully understood. Thus, propagated liver Mø showing features of Kupffer cells can likely serve as an indispensable tool for revealing the pathology of and beneficial therapies targeting Kupffer cells in liver fibrosis.

### 4.2. Involvement of Eph/Ephrin in the Residence of Kupffer Cells inside the Hepatic Sinusoid

To the best of our knowledge, it remains unclear why the cell bodies of Kupffer cells reside in the liver sinusoid without migrating to/residing in the space of Disse, similar to the hepatic stellate cells, and/or migrate to the outside of the liver via the bloodstream. It is well known that Eph/ephrin interaction interferes with the interaction of integrin and integrin ligands in the process of cell-cell and cell-matrix adhesion [12,13]. We previously showed that stimulation of certain EphAs and ephrin-As promotes cell adhesion through interactions with integrins and integrin ligands in monocytes [14]. Thus, we hypothesized that Ephs and ephrins are involved in the adhesion between Kupffer cells and LSECs via the interaction of integrins with their ligands. However, the expression patterns of Ephs and ephrins in Kupffer cells and LSECs, as well as that of integrins in Kupffer cells have not yet been comprehensively analyzed. The RT-PCR expression analyses of Ephs and ephrins revealed that propagated liver Mø express few member molecules, while primary LSECs expressed one or more member molecules in each of the EphA, ephrin-A, EphB, and ephrin-B classes. Moreover, our immunohistochemistry results demonstrated that Kupffer cells and LSECs express EphA2, ephrin-A1, EphB4, and ephrin-B1, which are representative molecules of each class among the molecules expressed in these cultured cells. This is the first comprehensive expression analysis of Ephs and ephrins in liver cells, and our findings indicate that EphA/ephrin-A and EphB/ephrin-B interactions likely arise when Kupffer cells are in contact with LSECs due to the promiscuous binding of Ephs to ephrins among the same classes [12,13]. This indication is partly supported by our Western blotting findings, in which EphA2 and EphB4 were tyrosine-phosphorylated in the liver, thus showing the contact of EphA2- and EphB4-bearing cells with ephrin-A- and ephrin-B-bearing cells, respectively. Moreover, by expression analyses of integrin subunits, we showed that propagated liver Mø likely express a substantial amount of α4β1 and large amounts of α5β1, α6β1, αLβ2, αMβ2, and αXβ2 integrins in the cell surface. The primary LSECs express large amounts of ICAM-1 and substantial amounts of VCAM-1, both of which are implicated in leukocyte adhesion in inflamed vascular endothelial cells [33]. It is well accepted that αLβ2, αMβ2, and αXβ2 bind to ICAM-1 and α4β1 binds to VCAM-1 [21,22]. Thus, we speculated that the adhesion of Kupffer cells to LSECs mainly arises from the binding of αLβ2, αMβ2, and αXβ2 with ICAM-1 at a steady state. It is well accepted that Eph/ephrin interaction interferes with integrin-integrin ligand interaction and generally induces cell-cell repulsion and cell migration by regulating the organization of the actin cytoskeleton, mainly through Rho-family GTPases, and rarely induces cell-cell adhesion [12,13]. Previously, we found that the EphB/ephrin-B interaction induced cell-cell repulsion in primary gastric epithelial cells [26]. We also found that the EphA/ephrin-A interaction promotes cell adhesion through interactions with integrins and integrin ligands in monocytes [14]. Based on present findings on the expressions of Ephs/ephrins and integrins/integrin ligands in propagated liver Mø and LSECs, we planned to examine whether EphA/ephrin-A and EphB/ephrin-B interactions promote or inhibit the adhesin of liver Mø to the mimicked LSEC surface on which the ephrin-A1, EphA2, ephrin-B1, or EphB4 protein was adsorbed in stripes and ICAM-1 was entirely overlaid. We found that liver Mø preferentially reside in not only the ICAM-1 plus ephrin-A1- or EphA2-adsorbed surface, but also the ICAM-1 plus ephrin-B1- or EphB4-adsorbed surface. Moreover, around the boundary, a blank zone without cells appeared only in ICAM-1-adsorbed control regions adjacent to ICAM-1 plus ephrin/Eph protein-adsorbed test regions, and the boundaries between the control and test regions were much clearer in the ephrin-A1 and EphA2 stripes than in the ephrin-B1 and EphB4 stripes. This indicates that the liver Mø may move on the integrin ligand-adsorbed surface and that the Eph/ephrin interaction inhibits their movement, causing them to remain on the Eph/ephrin-adsorbed surface, which implies stronger EphA/ephrin-A interaction than EphB/ephrin-B interaction. Thus, both EphA/ephrin-A and EphB/ephrin-B interactions promote the adhesion of Kupffer cells to LSECs, possibly through activation of the integrin/integrin ligand interaction, and are possibly implicated in the residence of Kupffer cells within the liver sinusoid. The findings of this study are summarized in Figure 11. Further studies are required to determine the downstream signaling molecules induced by the Eph/ephrin interaction to promote the adhesion of Kupffer cells by interaction with integrin/integrin ligands.

## 5. Conclusions

We successfully propagated mouse liver tissue-resident Mø using mixed culture with fibroblastic cells expressing molecules indispensable for forming the niche of Kupffer cells. The Mø marker expression patterns in propagated liver Mø were identical to those in Kupffer cells. We revealed (i) gene expression of certain Ephs and ephrins including EphA2, ephrin-A1, EphB4, and ephrin-B4 in propagated liver Mø and primary LSECs, (ii) immunohistochemical localization of Eph/ephrin member molecules indicating common expression in Kupffer cells and LSECs of the mouse liver, and (iii) surface expression of several integrin α and β subunits indicating expression of α4β1, αLβ2, αMβ2, and αXβ2 integrins in propagated liver Mø and that of their corresponding ligands ICAM-1 and VCAM-1 in primary LSECs. We also demonstrated that both EphA/ephrin-A and EphB/ephrin-B interactions promote liver Mø adhesion to the ICAM-1-adsorbed surface, which mimics that of LSECs. Thus, these interactions between Kupffer cells and LSECs may be implicated in the residence of Kupffer cells within the liver sinusoid. This study may lead to further studies on regulating the residence and regeneration of Kupffer cells in associated hepatic disorders, although further studies are required to confirm our findings.

## Figures and Tables

**Figure 1 biomedicines-10-03234-f001:**
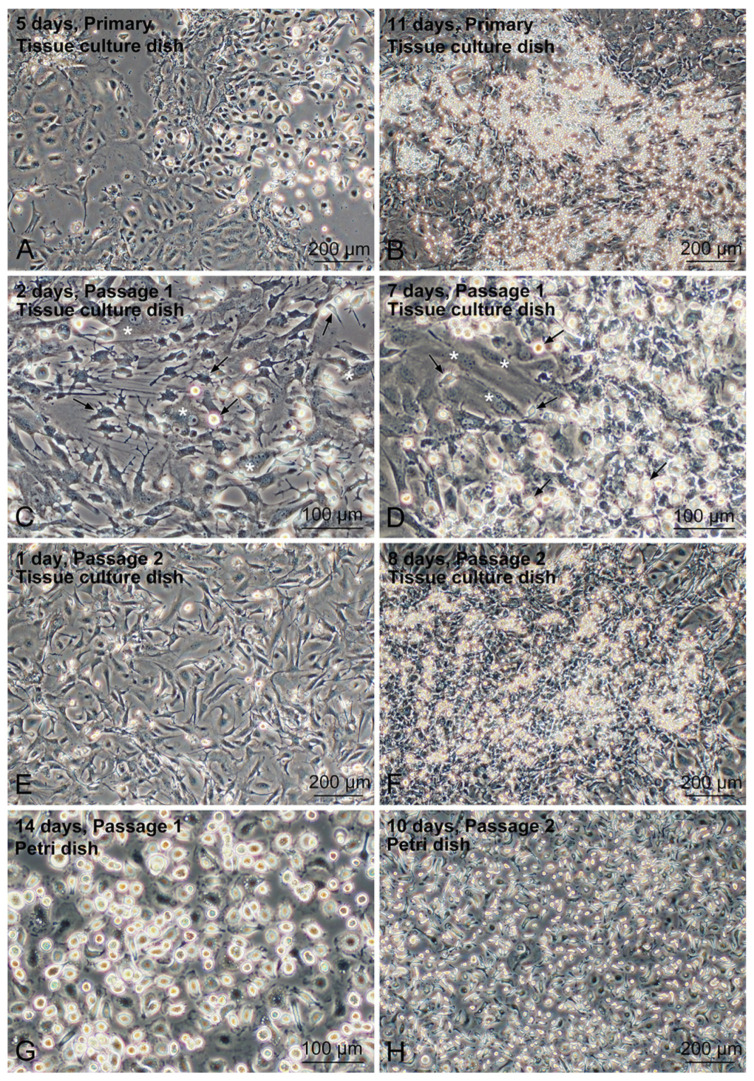
Phase-contrast micrographs showing the propagation of liver tissue-resident Mø in the mixed-culture as well as sub-culture with hepatic non-parenchymal cells in tissue-culture dishes and their segregation in bacteriological Petri dishes. (**A**,**B**) Primary liver cells were cultured for the indicated days after seeding in a tissue culture dish. (**C**,**D**) Liver cells after passage 1 were conducted on the indicated days after seeding in a tissue culture dish. The cells are mostly composed of non-parenchymal fibroblastic cells (*) and Mø (arrows). (**E**,**F**) Hepatic non-parenchymal cells after passage 2, on indicated days after seeding in a tissue culture dish. (**G**,**H**) Liver tissue-resident Mø in bacteriological Petri dishes. Nonadherent cells that formed cell aggregates were removed by washing with a conditioned medium. Mø selectively adhered to the surface of the dish.

**Figure 2 biomedicines-10-03234-f002:**
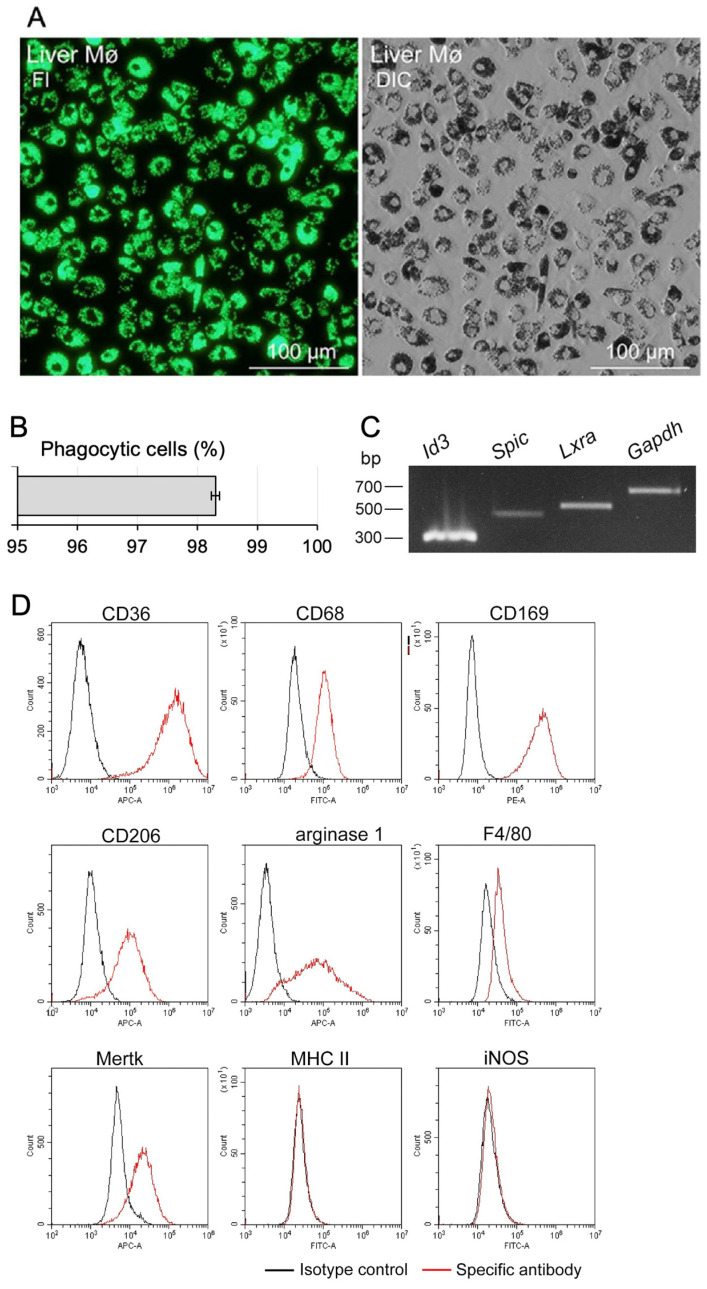
Propagated liver Mø using the mixed culture have identical properties to Kupffer cells. (**A**) Phagocytic properties assessed using incubation with fluorescent beads in liver tissue-resident Mø segregated by adhesion to bacteriological Petri dishes. Green fluorescence image (Fl, left panel) and differential interference images of the same fields (DIC, right panel). (**B**) Bar graph representing the percentage of propagated liver Mø in cells adherent to bacteriological Petri dishes, presented as mean ± SD. More than 690 cells per sample were counted, and the percentage of phagocytic cells in three mice was determined from three independent experiments. (**C**) Expression profile of transcription factors in propagated liver Mø determined using RT-PCR. Liver Mø clearly express *Id3*, *Spic*, and *Lxra* similar to Kupffer cells. (**D**) Representative histograms showing the expression profiles of nine Mø markers in propagated liver Mø using flow cytometry (red histogram, specific antibody; black histogram, isotype control). The cell suspensions were treated with a fluorochrome-labeled test antibody or the same amount of fluorochrome-labeled isotype control antibody. The characteristic expression patterns of CD206- and arginase 1-positive, as well as MHC II- and iNOS-negative, indicate polarization of propagated liver Mø to the M2 phenotype.

**Figure 3 biomedicines-10-03234-f003:**
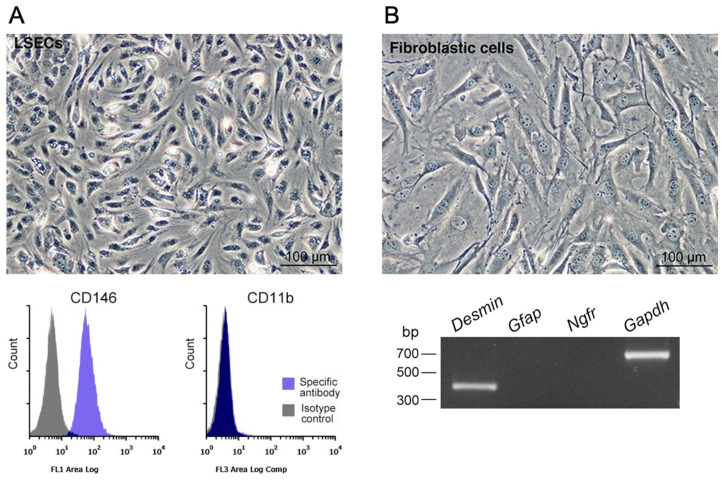
Phase-contrast images and expressions of specific markers in LSECs and fibroblastic cells propagated using mixed culture with liver Mø. (**A**) Representative phase-contrast image of primary LSECs showing small epithelial cell morphologies, and representative histograms showing CD146 and CD11b expression using flow cytometry (blue histogram, specific antibody; gray histogram, isotype control). CD146 is a specific marker of LSECs. (**B**) A representative phase-contrast image showing fibroblastic cell morphology and expression profile of hepatic stellate cell markers using RT-PCR in fibroblastic cell propagated using mixed culture with Mø from the liver.

**Figure 4 biomedicines-10-03234-f004:**
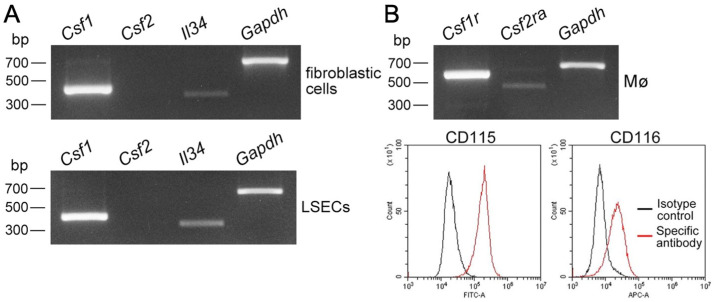
Expression of Mø growth factors in fibroblastic cells and LSECs as well as of their corresponding receptors in liver Mø propagated using mixed culture. (**A**) mRNA expression of Mø growth factors (*Csf1*, *Csf2*, *Il34*) amplified using RT-PCR in primary LSECs and fibroblastic cells propagated using mixed culture with liver Mø. Both cells express *Csf1* clearly and *Il34* substantially. (**B**) mRNA expression of Mø growth factor receptors (*Csf1r*, *Csf2ra*) amplified using RT-PCR and their cell surface expression (CSF1R, CD115; CSF2RA, CD116) determined using flow cytometry in propagated liver Mø in mixed culture (red line histogram, specific antibody; black line histogram, isotype control). Liver Mø express *Csf1r* clearly and *Csf2ra* substantially, while they clearly express CD115 and CD116. Liver Mø likely receive CSF1 and IL34 produced by LSECs and fibroblasts via CD115.

**Figure 5 biomedicines-10-03234-f005:**
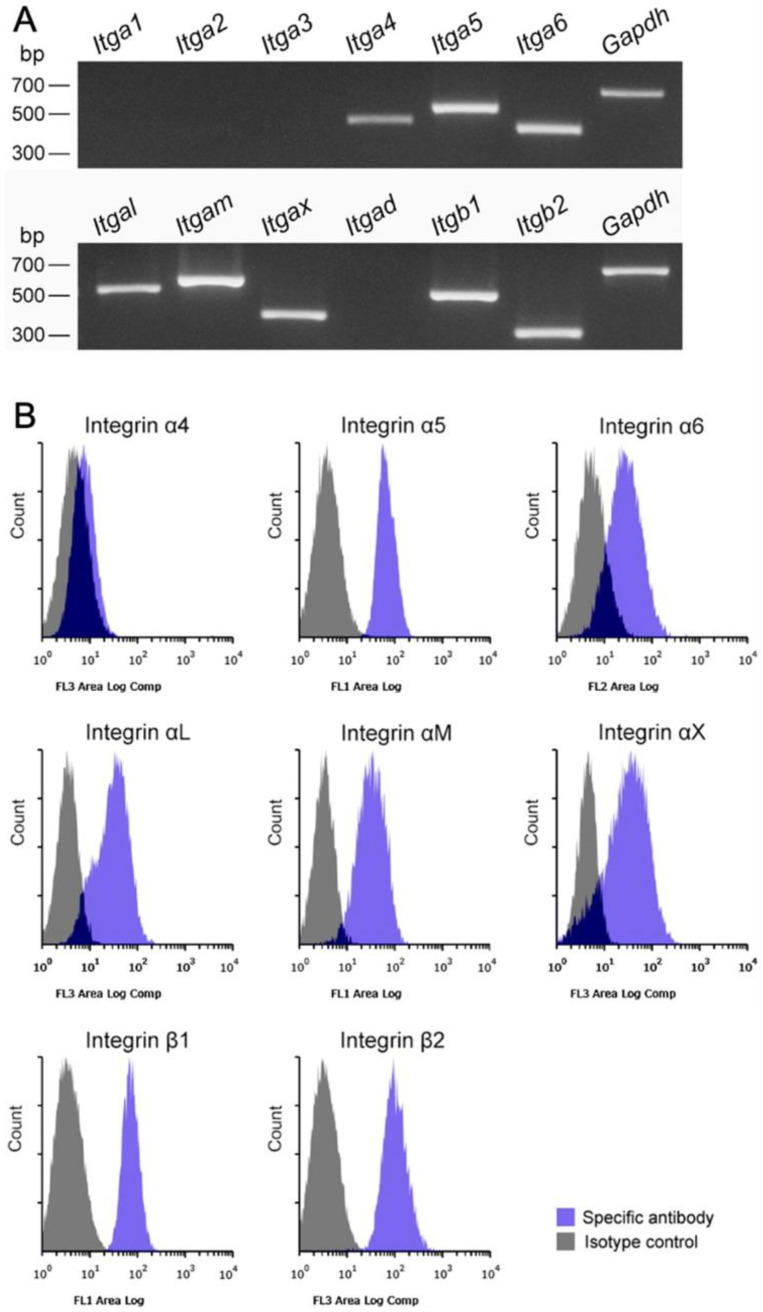
mRNA expression of integrin subunits and their cell surface expression in liver Mø propagated using mixed culture. (**A**) RT-PCR amplification of integrin subunits. Propagated liver Mø express *Itga4*, *Itga5*, *Itga6*, *Itgal*, *Itgam*, *Itgax*, *Itgb1*, and *Itgb2*. (**B**) Representative histograms analyzed using flow cytometry show cell surface expression of α4, α5, α6, αL, αM, αX, β1, and β2 integrin subunits in the liver Mø (blue histogram, specific antibody; gray histogram, isotype control). Liver Mø express relatively large amounts of α5β1, α6β1, αLβ2, αMβ2, and αXβ2 integrins, and substantial amounts of α4β1 integrin.

**Figure 6 biomedicines-10-03234-f006:**
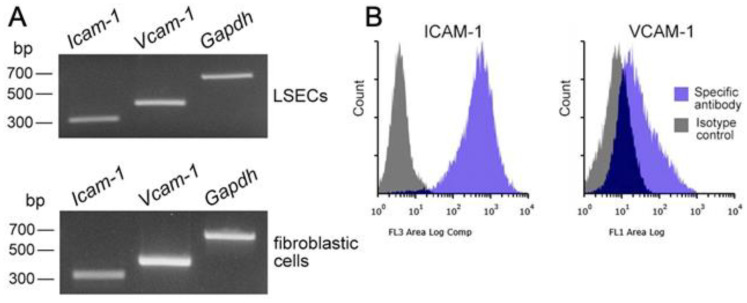
mRNA expression of integrin ligands in LSECs and fibroblastic cells, and their cell surface expression in LSECs. (**A**) mRNA expression of membrane proteins serving as integrin ligands (ICAM-1, VCAM-1) in primary LSECs and fibroblastic cells propagated using mixed culture with liver Mø. Both LSECs and fibroblasts clearly expressed *Icam-1* and *Vcam-1*. (**B**) Representative histograms analyzed using flow cytometry showing the cell surface expression of ICAM-1 and VCAM-1 in primary LSECs (blue histogram, specific antibody; gray histogram, isotype control). LSECs express ICAM-1 clearly while VCAM-1 is substantially expressed. Considering the findings in Figure 5, Kupffer cells likely adhere to LSECs via the β2 integrins/ICAM-1 and α4β1 and VCAM-1 interaction [21,22].

**Figure 7 biomedicines-10-03234-f007:**
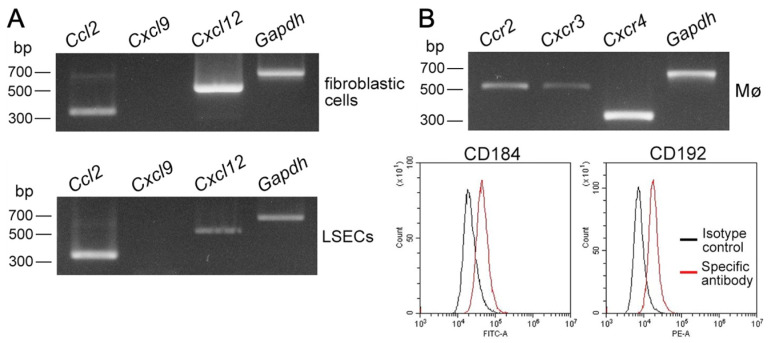
mRNA expression of chemokines in fibroblastic cells and LSECs, and cell surface expression of corresponding chemokine receptors in liver Mø propagated using mixed culture. (**A**) mRNA expression of chemokines (CCL2, CXCL9, CXCL12) in primary LSECs and propagated fibroblastic cells in mixed culture with liver Mø. Both cells clearly expressed *Ccl2* and *Cxcl12.* (**B**) mRNA expression of the corresponding chemokine receptors (CCR2, CXCR3, CXCR4) determined using RT-PCR and their cell surface expression (CD184, CXCR4; CD192, CCR2) determined using flow cytometry (red line histogram, specific antibody; black line histogram, isotype control) in propagated liver Mø in mixed culture. The propagated Mø expressed CD184 and CD192. Thus, liver Mø likely receive CCL2 and CXCL12 produced by LSECs and fibroblasts via CD192 and CD184, respectively.

**Figure 8 biomedicines-10-03234-f008:**
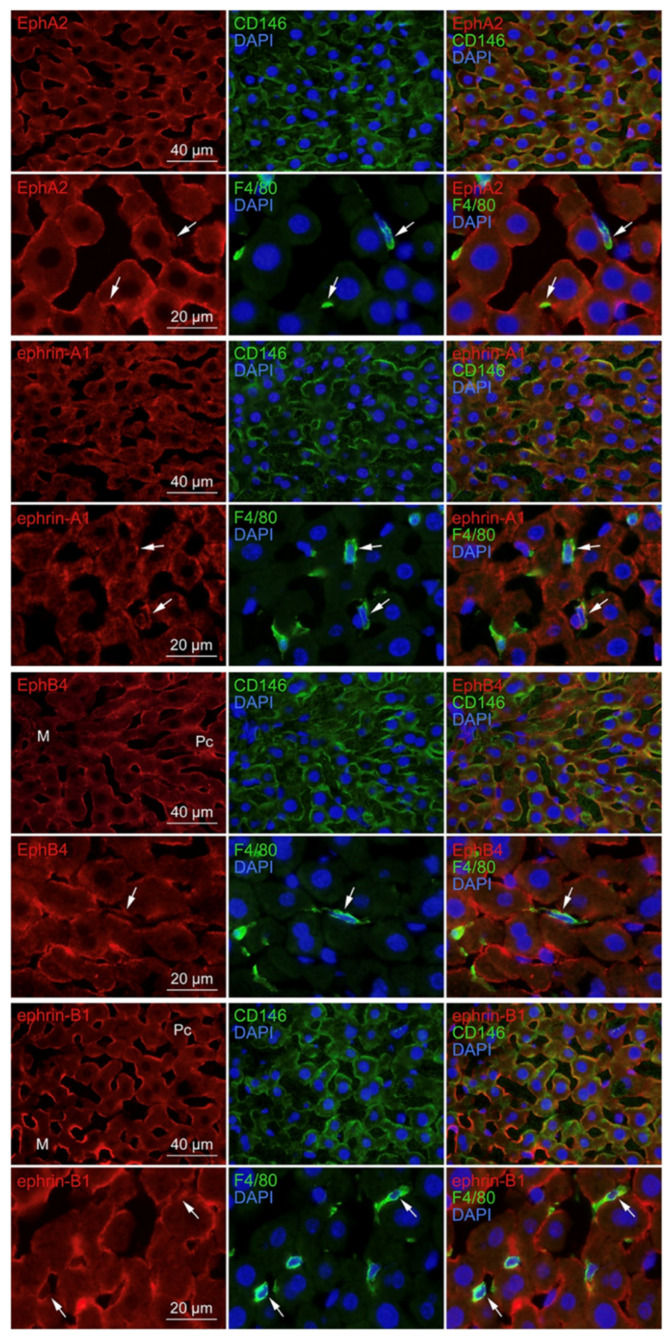
EphA2, ephrin-A1, EphB4, and ephrin-B1 localization in LSECs and Kupffer cells. Immunofluorescence micrographs showing EphA2, ephrin-A1, EphB4, and ephrin-B1 immunoreactivity (red) in CD146-positive LSECs (green) and F4/80-positive Kupffer cells (green) in the mouse liver. Cryostat sections were stained with the indicated antibodies and DAPI (blue). Green fluorescence images were merged with red, and/or blue fluorescence images in the same field. EphA2, ephrin-A1, EphB4, and ephrin-B1 immunoreactivity was localized in CD146-positive LSECs and weakly localized in F4/80-positive Kupffer cells (arrows). M, marginal region of the hepatic lobules; Pc, pericentral region of the hepatic lobules.

**Figure 9 biomedicines-10-03234-f009:**
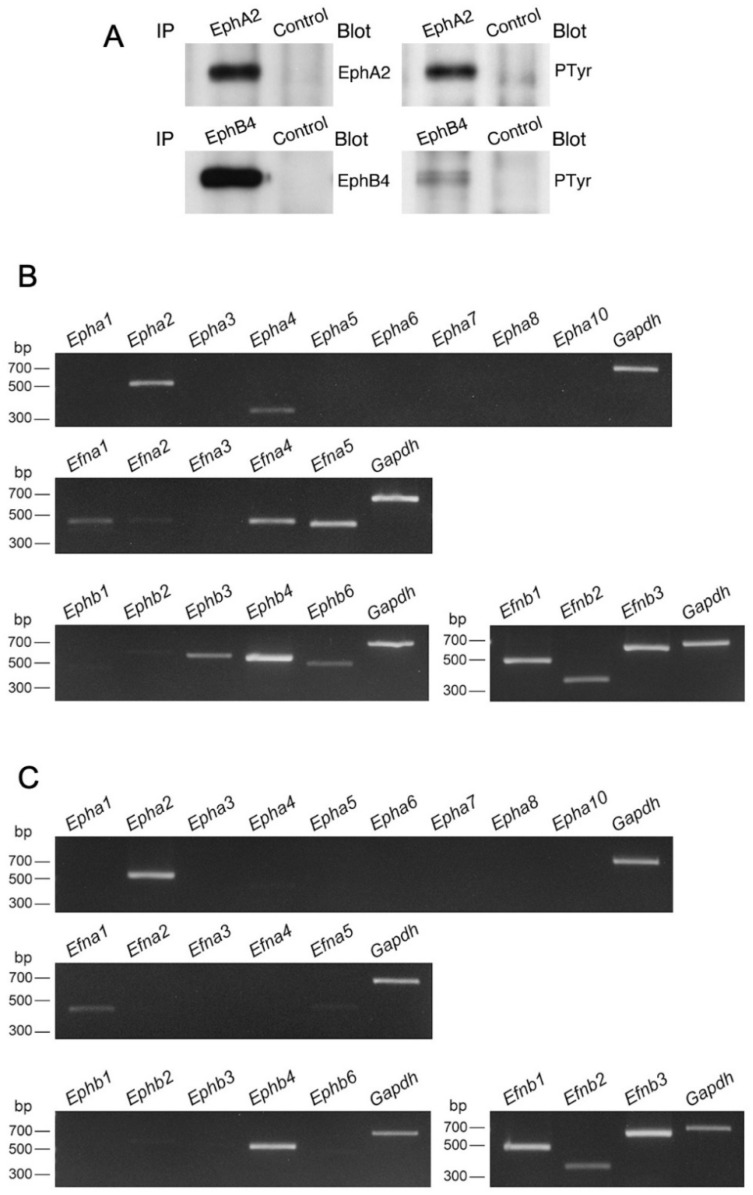
Expression of Ephs and ephrins in the mouse liver, liver Mø propagated using mixed culture, and LSECs. (**A**) The protein expression and tyrosine phosphorylation (PTyr) of EphA2 and EphB4 were detected using Western blotting in the mouse liver. EphA2 and EphB4 in the liver are highly and weakly tyrosine-phosphorylated, respectively. IP, immunoprecipitation. (**B**,**C**) RT-PCR analysis of all Ephs and ephrins in liver Mø propagated using mixed culture (**B**) and primary LSECs (**C**). Liver Mø express *Epha2*, *Epha4*, *Efna1*, *Efna4*, *Efna5, Ephb3*, *Ephb4*, *Ephb6*, *Efnb1*, *Efnb2*, and *Efnb3* whereas LSECs express *Epha2*, *Efna1*, *Ephb4*, *Efnb1*, *Efnb2*, and *Efnb3*.

**Figure 10 biomedicines-10-03234-f010:**
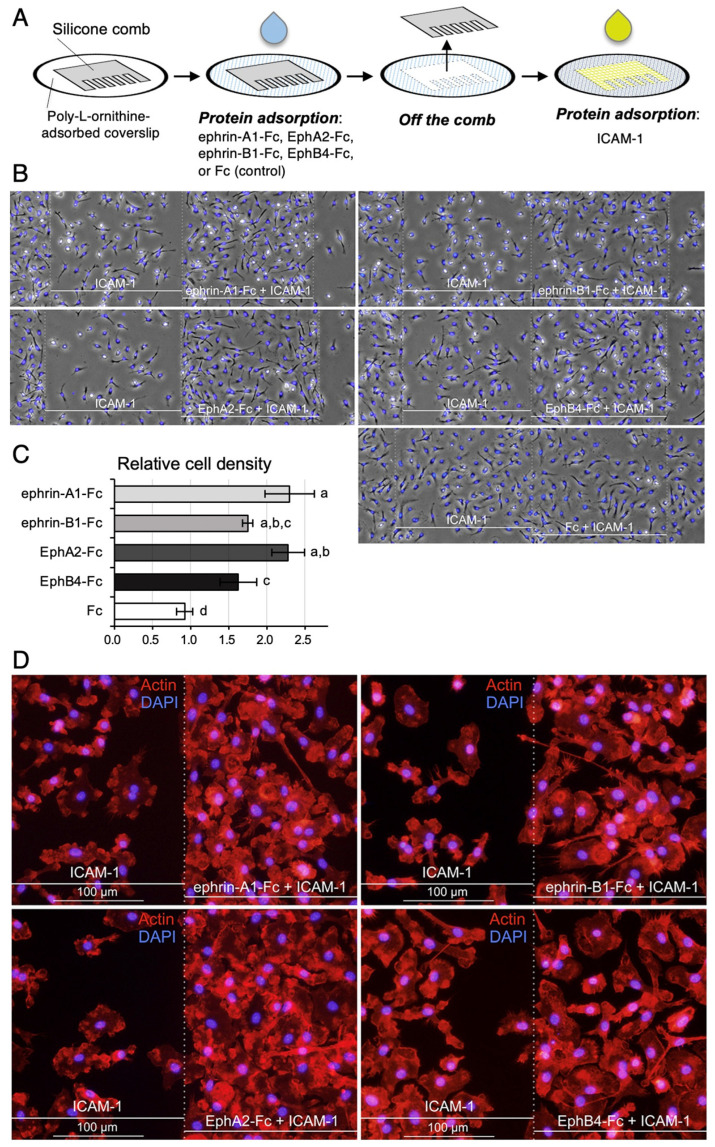
Liver Mø preferentially occupy the ephrin or Eph protein plus ICAM-1-adsorbed surface that mimics that of LSECs. (**A**) Schematic illustration of the ephrin or Eph protein adsorption procedure in stripes. The Fc chimera (test) or Fc protein (control) was adsorbed onto the coverslip surface, to which a comb-shaped silicon mask was applied. After washing and removing the mask, ICAM-1 was adsorbed onto the surface. The coverslips were placed in 6 cm culture dishes with culture medium. Liver Mø were plated and cultured for 16 h. Then, cells on the coverslips were fixed with Formalin/PBS and incubated with a mixture of Alexa 568-labeled phalloidin and DAPI. (**B**)Typical phase-contrast micrographs merged with the fluorescence images of DAPI staining (blue, nucleus) showing liver Mø cultured on the coverslip surface, wherein regions adsorbed with the indicated ephrin/Eph protein appeared as stripes. In the control Fc stripes, the liver Mø did not form stripes with different cell densities. (**C**) Quantified cell densities in regions adsorbed with ephrin/Eph or Fc protein plus ICAM-1 relative to those in adjacent control regions absorbed only with ICAM-1. Data from three independent experiments are shown as the mean ± SD. Values with different superscripts indicate significant differences among ephrin/Eph/Fc stripes (*p* < 0.05). (**D**) Typical fluorescence images of F-actin staining (orange-red) merged with those of DAPI staining showing the boundary between regions adsorbed with the indicated ephrin/Eph plus ICAM-1 and those with only ICAM-1 protein. Blank zones without cells appeared in the control region adjacent to the ephrin/Eph protein-adsorbed test region.

**Figure 11 biomedicines-10-03234-f011:**
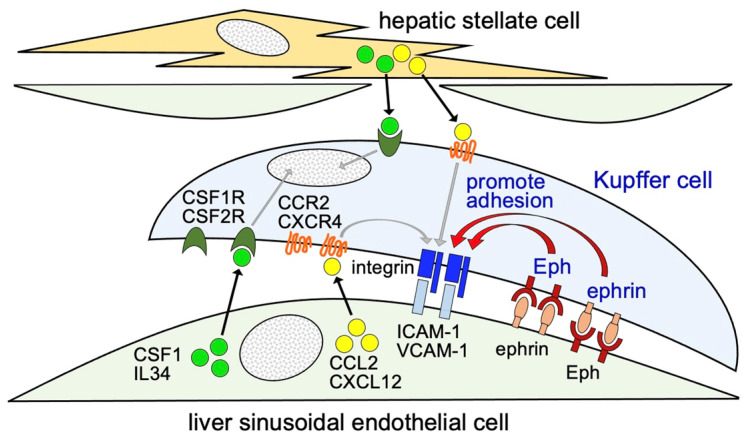
A schematic diagram highlighting the property of Kupffer cells colonizing the luminal surface of LSECs that is regulated by Eph/ephrin as well as certain integrins/integrin ligands, chemokines/chemokine receptors, and growth factors/growth factor receptors for Mø.

## Data Availability

The data presented in this study are available upon request from the corresponding author.

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
