# Peer review of "Eph/Ephrin Promotes the Adhesion of Liver Tissue-Resident Macrophages to a Mimicked Surface of Liver Sinusoidal Endothelial Cells"

_biomedicines, 2022, doi:10.3390/biomedicines10123234_

Round 1

Reviewer 1 Report

The work of Sho Kohara and Kazushige Ogawa using a mixed culture method, investigated whether Ephs and ephrins are involved in the adhesion of Kupffer cells to LSECs using propagated liver Mø.

They suggest that the Mø segregation method using the adhesion property of the bacteriological Petri dish is a reliable and simple purification method to obtain liver tissue-resident Mø from other non-parenchymal cells in mixed culture and they confirm by examination of  Id3, Spic, and LXRa. They further investigated mRNA expression of chemokines in fibroblastic cells and LSECs, and cell surface expression of corresponding chemokine receptors in liver Mø propagated by mixed culture.

Materials/Methods and Results are clearly presented.

Discussion well documented and the diagram highlighting the property of Kupffer cells colonizing the luminal surface of LSECs excellent.

Minor comments:

1.All chemokines data are by PCR and Flow cytometry. We agree for the transcriptional regulation of the chemokines but it would be of added value to confirm these results by ELISA.

2. Please indicate for IF and microscopy experiments the number of slides used for each experiment.

Author Response

Responses to the comments of Reviewer 1

We have revised the manuscript to address the comments of Reviewer 1 as outlined below. We have also highlighted the changes within the document in blue.

Comment 1: All chemokines data are by PCR and Flow cytometry. We agree for the transcriptional regulation of the chemokines but it would be of added value to confirm these results by ELISA.

Response: Thank you for your comment. We understand that additional data by ELISA improve reliability in expressions of the chemokines. However, the central theme of this study is whether Ephs/ephrins modulate adhesion via integrins/integrin ligands in liver macrophages, and we clearly showed the expressions of these proteins. Thus, we think that additional data on the chemokines do not essentially change the scientific value of the manuscript and that early publication of the manuscript is more important for readers of the journal.      

Comment 2: Please indicate for IF and microscopy experiments the number of slides used for each experiment.

Response: Thank you for your comment. According to your comment, we added the following sentence in Material and Methods “The immunohistochemical localization of those molecules was determined from three independent experiments using 42 liver sections derived from three mice. This is now on lines 264–266 of page 6.

Reviewer 2 Report

The article presented by Sho Kohara and Kazushige Ogawa, entitled “Eph/ephrin promotes the adhesion of liver tissue-resident macrophages to a mimicked surface of liver sinusoidal endothelial cells”, is an original article to assess to investigate the expression patterns of Ephs and ephrins, as well as their roles in resident macrophages (Kupper cells, restricted to the inside of the sinusoid). The methods used are well designed. The methodology selected to carry out the work is correct and adequate for the execution of the objectives set. The introduction is correct, contains the information necessary to focus the topic, and is up-to-date. In general, the work is well written and designed, and I only have small comments.

Minor revision:

1.       Line 367. Figure 1. The authors reach a series of conclusions with the images obtained in Figure 1. They differentiate the cell type by morphology. It would have been better to stain, but if not, it is suggested that the two types of cells described (macrophages and fibroblasts) be indicated by symbols within the image.

2.       In the discussion, it would be good if the authors related the findings to some clinical disease (hepatic fibrosis), in order to see the applicability.

Author Response

Responses to the comments of Reviewer 2

We have revised the manuscript to address the comments of Reviewer 2 as outlined below. We have also highlighted the changes within the document in blue.

Comment 1:  Line 367. Figure 1. The authors reach a series of conclusions with the images obtained in Figure 1. They differentiate the cell type by morphology. It would have been better to stain, but if not, it is suggested that the two types of cells described (macrophages and fibroblasts) be indicated by symbols within the image.

Response: According to your comment, we added asterisks and arrows in Fig. 1C and D to identify fibroblastic cells and macrophages, respectively, and accordingly, we changed a sentence in the caption of Figure 1. This is now on page 8.

Comment 2:  In the discussion, it would be good if the authors related the findings to some clinical disease (hepatic fibrosis), in order to see the applicability.

Response: According to the comment, we added some sentences describing the applicability of propagated liver Mø in the Discussion and used two more references. These are now on lines 618–623 of page 18 and the reference #31 and #32 in the reference list on page 22.